# Platelet Microparticles Protect Acute Myelogenous Leukemia Cells against Daunorubicin-Induced Apoptosis

**DOI:** 10.3390/cancers13081870

**Published:** 2021-04-14

**Authors:** Daniel Cacic, Håkon Reikvam, Oddmund Nordgård, Peter Meyer, Tor Hervig

**Affiliations:** 1Department of Hematology and Oncology, Stavanger University Hospital, 4068 Stavanger, Norway; oddmund.nordgard@sus.no (O.N.); peter.albert.meyer@sus.no (P.M.); 2Department of Clinical Science, University of Bergen, 5021 Bergen, Norway; hakon.reikvam@uib.no (H.R.); tor.audun.hervig@helse-fonna.no (T.H.); 3Department of Medicine, Haukeland University Hospital, 5021 Bergen, Norway; 4Department of Chemistry, Bioscience and Environmental Engineering, University of Stavanger, 4036 Stavanger, Norway; 5Laboratory of Immunology and Transfusion Medicine, Haugesund Hospital, 5528 Haugesund, Norway

**Keywords:** acute myelogenous leukemia, platelets, microparticles, apoptosis

## Abstract

**Simple Summary:**

Activated or apoptotic platelets both shed platelet microparticles that are proven to be internalized by many different cell types, including cancer cells. Here, we have investigated whether platelet microparticles can transfer their contents to the monocytic leukemia cell line THP-1 and if this could change cell activity and resistance to chemotherapy. We show that platelet microparticles were internalized by THP-1 cells and that platelet-associated microRNAs were elevated after a brief co-incubation. Furthermore, differentiation toward macrophages was induced and cell cycle progression, proliferation, and mitochondrial activity were decreased. Co-incubation with platelet microparticles increased chemotherapy resistance, which also was evident in acute myelogenous leukemia cells from patient samples, and it could be explained by the decrease in cell activity. Thus, platelet microparticles may have a role in the evolution of acute myelogenous leukemia and contribute to development of chemotherapy resistance, making them an interesting target for treatment.

**Abstract:**

The role of platelets in cancer development and progression is increasingly evident, and several platelet–cancer interactions have been discovered, including the uptake of platelet microparticles (PMPs) by cancer cells. PMPs inherit a myriad of proteins and small RNAs from the parental platelets, which in turn can be transferred to cancer cells following internalization. However, the exact effect this may have in acute myelogenous leukemia (AML) is unknown. In this study, we sought to investigate whether PMPs could transfer their contents to the THP-1 cell line and if this could change the biological behavior of the recipient cells. Using acridine orange stained PMPs, we demonstrated that PMPs were internalized by THP-1 cells, which resulted in increased levels of miR-125a, miR-125b, and miR-199. In addition, co-incubation with PMPs protected THP-1 and primary AML cells against daunorubicin-induced cell death. We also showed that PMPs impaired cell growth, partially inhibited cell cycle progression, decreased mitochondrial membrane potential, and induced differentiation toward macrophages in THP-1 cells. Our results suggest that this altering of cell phenotype, in combination with decrease in cell activity may offer resistance to daunorubicin-induced apoptosis, as serum starvation also yielded a lower frequency of dead and apoptotic cells when treated with daunorubicin.

## 1. Introduction

Acute myelogenous leukemia (AML) is a bone marrow malignancy originating in hematopoietic stem and progenitor cells [1,2,3]. The average 5-year survival rate for de novo disease is approximately 50% in younger patients [4], but this may vary widely depending on the occurrence of a selection of genetic aberrances. According to the 2017 European LeukemiaNet genetic risk stratification of AML, survival varies from 20% to over 60% [5]. Curative treatment involves intensive chemotherapy and, for select high-risk patient groups, the addition of consolidating treatment with allogenic stem cell transplantation, which carries the risk of a fatal outcome [6]. Thus, there is a need for a better understanding of tumorigenesis and evolution of the disease to improve treatment strategies.

Platelet–cancer interactions are becoming increasingly evident, and there is proof of cancer disease fundamentally altering the platelet transcriptome [7]. In aggregates with cancer cells, platelet function is hijacked to evade the NK cell response [8,9] and induce cancer cell epithelial–mesenchymal transition to facilitate metastasis [10,11]. Platelets are also important mediators for the development and maintenance of the cancer cell microenvironment [12,13].

Platelet microparticles (PMPs) are small membranous platelet particles (<1000 nm), which either bud off as a result of platelet activation [14] or as apoptotic bodies [15,16]. These microparticles are internalized by a variety of cell types, transferring their contents during this process [16,17,18,19]. PMPs contain a selection of the myriad of parent platelet alpha granule proteins [20,21] and platelet-associated microRNAs [22,23], which may potentially affect the biological behavior of the cells that have internalized them. This transfer of microRNAs has been demonstrated in a number of cancer models [24,25,26], where, although the effects are dependent on the cancer type and model, the PMPs appear to have both pro and anti-tumoral properties.

Targeting anti-apoptotic proteins is a novel strategy in the treatment of AML [27]. BCL2 is an important regulator of the intrinsic or mitochondrial apoptosis pathway, inhibiting BCL2 Antagonist/Killer (BAK) and BCL2 Associated X, Apoptosis Regulator (BAX) oligomerization, thus preventing pore formation in the outer mitochondrial membrane and subsequently leading to the leakage of cytochrome c and activation of caspase-9 [28]. Both platelet releasate and lysate seem to counter the effects of agents that specifically target this pathway, revealing an anti-apoptotic potential of platelets in AML [29]. There is also evidence that intrinsic apoptosis can be affected by the transfection of certain microRNAs, which are also found to be overexpressed in AML and present in platelets, indicating the potential relevance of these regulatory RNA molecules in an interaction between AML cells and platelets [30,31].

The role of microRNAs in AML is further supported by several studies showing an association of microRNA expression with mortality and chemotherapy resistance in patients, in whom several of the microRNAs are known to be present in high concentrations in platelets and platelet microparticles [32,33,34]. In this study, we aimed to assess whether PMPs could be taken up by AML cells and if this would change the AML cells’ microRNA levels and in vitro chemotherapy resistance.

## 2. Results

### 2.1. Platelet-Associated microRNAs Are Increased in THP-1 Cells after PMP Co-Incubation

To examine whether platelet microparticles could be internalized by AML cells, we cultured cells from the monocytic AML cell line THP-1, with acridine orange (AO)-stained PMPs for 18 h. There was a PMP concentration-dependent increase of fluorescence in the co-incubated cells, and fluorescence microscopy revealed that that the stain was indeed dispersed within the cell nucleus and not located to bound microparticles (Figure 1A,B). To further investigate whether this PMP internalization could increase microRNA levels, we analyzed a selection of platelet-associated microRNAs in THP-1 cells after 18 h of co-incubation with PMPs. miR-125a-5p, miR-125b-5p, and miR-199-5p levels were all markedly increased (Figure 1C). This was particularly true for miR-199-5p, where levels were undetectable without PMP co-incubation in 2/3 samples (range 37.55–not detected), versus an average Cq value of 33.20 (range 33.13–33.24) with PMP co-incubation. These findings give indirect proof that microRNAs can be transferred from platelets to THP-1 cells through PMP internalization.

### 2.2. PMPs Lead to Increased Resistance of THP-1 Cells to DNR

Both miR-125a-5p and miR-125b-5p have been associated with resistance to chemotherapy in retroviral transduction studies [30,31]. Therefore, we examined whether the cytotoxic effect of daunorubicin (DNR), a common front-line chemotherapeutic in AML, could be influenced by PMP internalization. Co-incubation of THP-1 cells with PMPs decreased the relative frequency of dead and apoptotic cells in a concentration-dependent manner following treatment with DNR (Figure 2A). Thus, for all other analyses, PMPs were co-incubated at a concentration of 1.5 × 10^7^ per mL medium, unless otherwise specified, as this generated the highest chemoprotective effect. Vector control experiments, where the supernatant of isolated PMPs was added to Iscove’s Modified Dulbecco’s Medium (IMDM) + 10% FBS medium at a concentration of 5%, only had a small and non-significant effect on resistance to DNR (*p* = 0.109).

### 2.3. Co-Incubation of Primary AML Cells with PMPs Also Increased Resistance to DNR

As there are limitations for the clinical relevance of cell line AML models [35], we examined whether the chemoprotective effect of PMPs could be observed in cells derived from AML patients. Using a similar approach, albeit with serum-free conditions, our results showed an identical effect on primary AML cells where we identified a significantly lower frequency of dead and apoptotic cells when PMPs were added (Figure 2B). The average absolute reduction in dead and apoptotic cells in individual patient samples ranged from 0.2 to 55.9% and was significant in 8/9 patients (Appendix A). One-way ANOVA analysis of the effects on THP-1 cells of PMPs from different releasates (when used in the primary AML experiments as quality controls) revealed no significant inter-releasate batch difference (*p* = 0.823).

### 2.4. THP-1 Cells Co-Incubated with PMPs Had Lower Caspase-9 Activity Following DNR-Treatment

Using microRNA databases (miRDB [36] and TargetScanHuman [37]), we found several predicted target mRNAs for miR-125a-5p and miR-125b-5p with important roles in the intrinsic apoptotic pathway, such as the pro-apoptotic BCL2 family proteins, BCL2 Modifying Factor (BMF) and BAK1 [38]. To investigate whether PMP internalization could influence the intrinsic apoptotic pathway, THP-1 cells were co-incubated with PMPs and treated with DNR using the established approach. Then, the cells were analyzed for caspase-9 activation, which is a downstream effect of mitochondrial outer membrane permeabilization (MOMP). There was a lower frequency of caspase-9 positive cells in the THP-1 cell cultures co-incubated with PMPs (Figure 3), suggesting that the chemoprotective effect of PMPs could be the result of the effects on the intrinsic apoptotic pathway upstream of caspase-9 activation.

### 2.5. Decreased Cell Activity Protected THP-1 Cells against DNR

Cytotoxic chemotherapy is believed to be most potent in highly proliferating cancer cells. Inducing cellular dormancy to decrease DNA replication should theoretically offer a chemoprotective effect, as it would prevent DNR-triggered DNA damage. Thus, we investigated whether serum starvation would decrease cell activity and subsequently protect THP-1 cells against DNR. We analyzed proliferation, cell cycle distribution, and mitochondrial membrane potential, and we observed that 48 h of serum starvation in THP-1 cells induced a significant growth arrest (Figure 4A–C). To evaluate whether serum starvation affected DNR-resistance, we compared apoptosis and cell death in THP-1 cells, with or without serum starvation, 24 h after treatment with 0.5 µM DNR. We showed a marked reduction in the frequency of dead and apoptotic cells (Figure 4D).

To investigate whether the apparent chemoprotective effect of PMP co-incubation may be the result of a similar decrease in cell activity, we measured the effects of PMPs on cell proliferation, cell cycle distribution, and mitochondrial membrane potential. Our results showed that co-incubation with PMPs increased the frequency of cells in the G0/G1 cell phase, reduced mitochondrial membrane potential, and inhibited cell proliferation (Figure 5A–C). These findings lead us to believe that PMPs may protect THP-1 cells from DNR-induced cell death by partially inhibiting cell cycle progression and proliferation. Co-incubation with PMPs did not alter mRNA or protein levels of CDK4 (Figure 5D,E), which is fundamental for THP-1 viability and normal cell cycle progression [39,40].

### 2.6. PMP Co-Incubation Increased Differentiation of THP-1 Cells toward Macrophages

THP-1 cells are capable of macrophage differentiation, leading to cell growth arrest. Thus, we wanted to examine if increased differentiation of the cells could contribute to the observed decrease in cell cycle progression. PMP co-incubation increased both forward scatter and side scatter (Figure 6A), indicating increased cell size and granularity, which are two hallmarks of macrophage differentiation [41]. Surprisingly, we could not corroborate the forward scatter findings with measurement of cell cross-sectional area using the particle analysis function in the ImageJ software with pictures taken under an inverted phase-contrast microscope (Figure 6B). However, co-incubation with PMPs led to a significant increase in CD14 antigen expression (Figure 6C). Thus, the decrease in cell cycle progression, and therefore part of the chemoprotective effect, could stem from differentiation of the cells toward macrophages.

## 3. Discussion

The important role of platelets in cancer development, progression, and metastasis is becoming increasingly clear, and there are several known mechanisms in this interplay, including the transfer and uptake of platelet microparticles. We showed that co-incubating primary AML cells and THP-1 cells with PMPs increased their DNR resistance. In addition we demonstrated the inhibition of THP-1 cell proliferation, cell cycle progression, mitochondrial membrane potential, and induction of differentiation toward macrophages.

We observed internalization of PMPs by THP-1 cells and a subsequent increase of platelet-associated microRNAs. The internalization was observed using AO-stained PMPs. AO will stain DNA, RNA, and acidic vesicles. Platelets do not contain DNA, other than a small amount of mtDNA. However, platelets and PMPs both contain RNA and lysozomes, making AO staining suitable for our purpose.

Co-incubation of THP-1 or primary AML cells with PMPs increased their resistance to DNR-treatment at a concentration of 0.5 µM, which is close to the peak plasma concentration measured in patients receiving 60 mg/m^2^ DNR [42]. In AML, DNR is routinely administrated at dosages of 45–90 mg/m^2^ [43]. This observed chemoprotective effect could be the result of microRNA transfer from PMPs. The overexpression of both miR-125a and miR-125b have been associated with DNR resistance in AML cell lines, including THP-1 [30,31]. Using the transduction of THP-1 cells with murine stem cell virus (MSCV), these studies obtained a 4 and 4.9-fold increase of miR-125a and miR-125b. In our study, the respective levels were increased 3.0 and 3.2-fold. The transduction of miR-125a and miR-125b resulted in downregulation of the apoptotic proteins Grk2 and Puma [30,31]. The latter is a member of the pro-apoptotic BH3-only proteins of the intrinsic apoptosis pathway [38], where we show a decrease in activation with PMP co-incubation. These proteins are known to be crucial participants in apoptosis, as genetic knockout models are protected against several apoptotic stimuli [44]. However, the cited studies with the transduction of miR-125a and miR-125b did not include an analysis of cell activity in the THP-1 cells associated with microRNA overexpression, which may be decisive for the actual DNA-damage induced by DNR. Other studies have linked the ectopic expression of miR-125a with proliferation inhibition, although in solid tumor cell lines [45,46].

PMPs contain several hundred different proteins and small RNAs, meaning the underlying mechanism for chemoprotection are likely more complex than that reflected in the single microRNA transduction studies. The anti-cancer effect of DNR and other anthracyclines are believed to mainly be a result of interference with topoisomerase II (Top2) enzyme activity [47]; however, other mechanisms have been identified [48]. Top2 introduces double-strand DNA breaks during replication [47]; thus, an inhibition of proliferation should decrease the efficiency of Top2 poisons. We showed that decrease of cell activity through serum starvation protects THP-1 cells against DNR-triggered apoptosis and cell death, and we suggest that PMPs could offer chemoprotection through this mechanism.

Vasina and colleagues have previously shown that microparticles from apoptotic platelets can induce macrophage differentiation in THP-1 cells after 7 days of co-incubation [16]. Here, we show prominent upregulation of CD14 antigen already after 48 h using platelet microparticles from activated platelets from platelet concentrates containing a mixture of PMPs generated by activation and apoptosis, better resembling the in vivo milieu. Seemingly, there are conflicting results regarding cell size analyses, as forward scatter and side scatter were increased, but the measured cross-sectional cell area was unchanged. However, we believe the increase in light scatter was affected by morphological changes with more vacuolization in the cells treated with PMPs.

The observed differentiation effect can at least partially explain the decrease in cell cycle progression, as THP-1 cells treated with phorbol myristate acetate for macrophage differentiation only exit G1 phase to a little extent [49]. CDK4 mRNA and protein levels were unchanged, and the decrease in cell cycle progression would appear to be the result of a downstream target. However, notable downregulation of the CDK4 gene is known to be a later event in macrophage differentiation of THP-1 cells [49]. We have not identified the exact substances that initiate differentiation or lead to the inhibition of cell cycle progression. The latter could partially be an independent process, because contrary to our findings, mitochondrial activity is increased with macrophage differentiation [41]. Transforming growth factor beta (TGF-β) is a potent cell cycle regulator known to be present in platelets. TGF-β induces dormancy or quiescence through several mechanisms, but it cannot be entirely responsible for the observed chemoprotective effect of PMPs, as it is known to be abundant in the platelet secretome [50,51], and we did not observe any significant effects on the frequency of dead and apoptotic cells in our vector control experiments.

Several research groups have reported that PMPs affect cancer development. Michael and colleagues showed that PMPs could infiltrate solid tumors and inhibit the growth of lung and colon cancer [26]. Others have linked PMPs to increased epithelial–mesenchymal transition and metastatic capacity in ovarian cancer [25] and lung cancer invasion [24]. Recent evidence has shown that platelets can have a bimodal effect in colorectal cancer where they inhibit growth but promote metastasis [52]. An extensive review of the role of PMPs in cancer progression can be found elsewhere [53].

Our protocol for the quantitation of microparticles has some limitations, as the PMP number per mL releasate varied on average by 10.0%, but it ranged from 0.4 to 34.9% between technical replicates. Thus, the final concentration of PMPs in the culture media may have varied extensively in some experiments. Accordingly, the concentration of microRNAs and proteins will vary from batch to batch of platelet concentrates. We accounted for the latter when we chose to use pooled platelet concentrates derived from four different donors. Furthermore, we found no inter-batch differences with respect to chemoprotective effect.

The induction of resistance to DNR by PMPs could have significant clinical relevance. Inhibiting the production of PMPs may present a potential therapeutic approach in AML to increase chemosensitivity. This can easily be achieved with platelet inhibition [54]. Platelet inhibition has also previously been linked to both lower cancer incidence and improved cancer-specific survival [55,56,57,58], although the exact mechanism is unknown. On the flip side, the differentiation of AML cells by PMPs might be beneficial to inhibit evolution of the disease.

## 4. Materials and Methods

### 4.1. Cell Line

The THP-1 cell line was purchased from ATCC (American Type Culture Collection; Manassas, VA, USA) and maintained in Iscove’s Modified Dulbecco’s Medium (IMDM; Thermo Fisher Scientific, Waltham, MA, USA) + 10% FBS (Sigma Aldrich, St. Louis, MO, USA). Only cells in the exponential growth phase were used, and cultures were kept for less than three months.

### 4.2. Primary AML Cells

Primary AML cells were isolated by density gradient separation of peripheral blood from consenting patients at the Department of Medicine, Section of Hematology, Haukeland University Hospital (Bergen, Norway). The cells were cryopreserved in liquid nitrogen until use. The cryopreservation solution consisted of insulin-free RPMI 1640 (Sigma Aldrich), supplemented with 10% dimethylsulfoxide and 20% FBS. Primary AML cells were cultured in StemSpan Serum-Free Expansion Medium (Stem Cell Technologies, Vancouver, BC, Canada) with the addition of the following recombinant cytokines in a final concentration of 20 ng/mL: stem cell factor (Peprotech EC, London, UK), G-CSF (Peprotech), and FMS-like tyrosine kinase 3 ligand (Peprotech). Charateristics of the AML patients can be found in Table 1.

### 4.3. Platelet Concentrate

Routinely prepared platelet concentrates pooled from four donors (Tacsi system; Terumo BCT, Lakewood, CO, USA) were provided by the Department of Immunology and Transfusion Medicine, Stavanger University Hospital (Stavanger, Norway), after written consent from the donors. The platelet concentrations were 0.88–1.08 × 10^9^ per mL. Leukocytes were removed by filtration to a residual level of <1.00 × 10^6^. In the final concentrate, the storage medium contained approximately 65% additive solution (PAS-III, Baxter, Lake Zurich, IL, USA) and 35% plasma.

### 4.4. Platelet Releasate

The platelet concentrate was transferred from the blood bag to separate 50 mL tubes and incubated with a final concentration of 1 U/mL human thrombin (Sigma Aldrich) for one hour in a 37 °C water bath. The tubes were gently agitated every 5 min. The platelet releasate was centrifuged for 10 min at 900× *g*, and the supernatant was transferred to new 50 mL tubes. The samples were stored at −80 °C. Fibrin clots that appeared after thawing were plucked using a 10 mL serological pipette.

### 4.5. Platelet Microparticles Isolation, Co-Culture, and Measurement

Platelet releasate was centrifuged at 15,000× *g* for 90 min at room temperature and the supernatant was carefully poured off. To examine the biological effects of platelet microparticles, a mastermix of StemSpan + cytokines (for primary cells), or IMDM + 10% FBS (for THP-1 cells), was used to resuspend PMPs before transfer to cell culture and thoroughly mixed with the cells by pipetting. Two hours after the PMPs were added to the cell cultures, the wells were mixed again by pipetting. For quantitation, the microparticles were resuspended in 400 µL of 0.22 µm filtered Annexin V Binding Buffer (Miltenyi Biotec, Bergisch Gladbach, Germany), before 200 µL of the solution was transferred to a second tube. Twenty µL of Annexin V FITC (Milteny Biotec), and 2 µL of anti-CD61 APC (clone Y2/51; Miltenyi Biotec), or 22 µL of 0.22 µm filtered Annexin V Binding Buffer for an unstained control, were added and incubated for 15 min at room temperature. Finally, 278 µL of 0.22 µm filtered Annexin V Binding Buffer and 50 µL CountBright beads (Thermo Fisher Scientific) were added before analysis. Microparticle gates were set with Megamix-PLUS FSC beads (size range of beads: 0.3 to 0.9 µm; BioCytex, Marseille, France) using the side scatter channel, according to Poncelet and colleagues [59]. At least 2500 bead events were collected. This as well as all other flow cytometric analyses were performed on a CytoFLEX flow cytometer (Beckman Coulter, Brea, CA, USA) using CytExpert ver. 2.4 acquisition and analysis software (Beckman Coulter).

### 4.6. Acridine Orange Staining of Platelet Microparticles

Platelet releasate was stained with 100 µg/mL of acridine orange (Thermo Fisher Scientific) and incubated for 30 min at room temperature. Then, the solution was washed and centrifuged two times at 15,000× *g* for 90 min. Tubes were changed after the first wash step to avoid any contamination of acridine orange that may have adhered to the plastic. Finally, the PMPs were resuspended in IMDM + 10% FBS and co-cultured with THP-1 cells for 18 h at a concentration of 5 × 10^6^ PMPs per mL. The cells were harvested and washed twice in Dulbecco’s phosphate-buffered saline (DPBS; Sigma Aldrich) before analysis with flow cytometry using the FITC channel, and with a Zeiss Axioplan 2ie MOT fluorescence microscope (Carl Zeiss, Göttingen, Germany) using an SpGreen filter. At least 25,000 gated cells were collected for flow cytometric analysis.

### 4.7. mRNA and microRNA Analysis

Total RNA was isolated using the miRNeasy kit (QIAGEN GmbH, Hilden, Germany), and RNA concentration was measured on a NanoDrop 2000 spectrophotometer (Thermo Fisher Scientific). Reverse transcription was performed with the TaqMan MicroRNA Reverse Transcription Kit (Thermo Fisher Scientific), and the High-Capacity cDNA Reverse Transcription Kit (Thermo Fisher Scientific). Real-Time PCR was done on the Mx3005P qPCR system (Agilent Technologies, Palo Alto, CA, USA) using TaqMan MicroRNA Assays (Thermo Fisher Scientific) with the TaqMan Universal Master Mix for microRNA analyses (Thermo Fisher Scientific), and the TaqMan gene expression assays (Thermo Fisher Scientific) with the TaqMan Gene Expression Master Mix (Thermo Fisher Scientific) for mRNA analyses, following the manufacturer’s instructions. *BCR* or *ACTB* were used as reference genes, and the relative expression was calculated using the 2^−ΔΔCq^ method. ΔΔCq was calculated as ΔCq value (target gene minus reference gene) for cells without PMP co-incubation minus ΔCq value for PMP co-incubated cells. For a comprehensive list of the microRNA and mRNA assays used in this study, see Appendix A.

### 4.8. Daunorubicin Apoptosis Assay

Approximately 5 × 10^5^ cells per mL of resuscitated primary AML cells, or THP-1 cells in exponential growth phase, were cultured under aforementioned conditions with or without PMPs for 24 h. The cells were then treated with 0.5 µM daunorubicin hydrochloride (Sigma Aldrich) for another 24 h before further analysis. THP-1 cells were also used as a quality control for the experiments with primary AML cells. Then they were kept in the same batch of StemSpan + cytokines to detect any false negative results in case of issues with the PMP isolation. Cell viability was analyzed with the Annexin V-FITC kit (Miltenyi Biotec), strictly following the manufacturer’s instructions. Dead and apoptotic cells were analyzed using flow cytometry and gated out in a single gate using a pseudo color plot of FITC-A versus PerCP Cy 5.5-A after doublet discrimination. For analysis with primary AML cells, contaminating cells were gated out based on light scatter properties. See Appendix A for gating strategies. At least 25,000 gated cells were collected.

### 4.9. Caspase-9 Activity

Caspase-9 activity in daunorubicin-treated THP-1 cells was measured using the CaspGLOW Fluorescein Active Caspase-9 Staining Kit (Thermo Fisher Scientific). Approximately 5 × 10^5^ cells in 0.3 mL IMDM + 10% FBS were stained with 1 µL FITC-LEHD-FMK and incubated for 30 min in a CO_2_ incubator before washing twice with the supplied wash medium and analysis with flow cytometry. Both untreated and treated, but not stained THP-1 cells, were used as negative controls. At least 25,000 gated cells were collected.

### 4.10. Mitochondrial Membrane Potential

Mitochondrial membrane potential was assessed using the MitoProbe DiIC1(5) Assay Kit (Thermo Fisher Scientific). THP-1 cells were cultured with or without PMPs for 24 h before 5 × 10^5^ cells in 1 mL IMDM + 10% FBS were stained with DiIC1(5) using carbonyl cyanide 3-chlorophenylhydrazone-treated cells as a correction for background signal and incubated for 30 min following the manufacturer’s instructions. After doublet discrimination, MFI (mean fluorescence intensity) values of gated cells were compared using the APC channel on the flow cytometer. See Appendix A for gating strategy. At least 30,000 gated cells were collected.

### 4.11. Cell Cycle Analysis

THP-1 cells were incubated for 24 h in IMDM + 10% FBS with or without PMPs. Cells were washed and 5 × 10^5^ cells were stained with 10 µM Vybrant Dye Cycle Green Stain (Thermo Fisher Scientific) and incubated for 30 min in a 37 °C water bath. Immediately after incubation, the cells were analyzed using the FITC channel on the flow cytometer. 2N cells, representing G0/G1 cell phase, were gated out after doublet discrimination. See Appendix A for gating strategy. At least 10,000 gated cells were collected.

### 4.12. Flow Cytometry Proliferation Analysis

Proliferation analysis was performed with the Cell Trace Far Red Proliferation Kit (Thermo Fisher Scientific) and analyzed with the APC channel on the flow cytometer. On day 0, THP-1 cells at a concentration of 1 × 10^6^ per mL were stained with 5 µM Far Red reagent in DPBS and incubated briefly for 5 min in a 37 °C water bath to avoid excessive cell toxicity. The stained cells were washed with IMDM + 20% FBS and cultured as previously described. Medium with or without PMPs was added on days 2 and 4 to keep concentration of cells below 8 × 10^5^ per mL. A sample of the cells was analyzed on day 0 to identify baseline MFI. At least 25,000 gated cells were collected.

### 4.13. Flow Cytometry Immunophenotyping

THP-1 cells were cultured under aforementioned conditions with or without PMP co-incubation and harvested after 48 h. Approximately 1 × 10^6^ cells were washed in DPBS, resuspended in 98 µL of DPBS containing 0.5% BSA, and labeled with 2 µL of anti-CD14 APC (clone REA599; Miltenyi Biotec). The cells were incubated for 10 min at 4 °C and washed before analysis. An unstained sample was used to determine background signal. At least 30,000 gated cells were collected.

### 4.14. Measurement of CDK4 by Indirect Intracellular Flow Cytomtery

THP-1 cells were cultured for 24 h before harvest and analyzed for intracellular protein using the published protocol by Ludwig and colleagues [60]. Briefly, cells were fixed and permeabilized using the eBioscience Foxp3/Transcription Factor Staining Buffer Set (Thermo Fisher Scientific). Then, cells were incubated with unconjugated anti-CDK4 (clone DCS-31; Thermo Fisher Scientific) and labeled with the proper conjugated secondary antibody. Dilution and incubation time can be found in Appendix A. A “no primary antibody” sample was used to subtract background signal. At least 25,000 gated cells were collected.

### 4.15. Measurement of Cell Cross-Sectional Area

For measurement of the cross-sectional area, cultured cells were transferred to a Bürker chamber to minimize the physical cell membrane manipulation and assessed under an inverted phase-contrast microscope. Four representative fields per technical replicate at 100× magnification were captured using an Olympus Pen Lite E-PL5 camera (Olympus, Tokyo, Japan). Pictures were analyzed using the particle analysis function of the ImageJ software ver. 1.52k [61]. Image optimization and thresholding was performed as described in the Appendix A.

### 4.16. Serum Starvation

In separate experiments, analysis of daunorubicin-induced apoptosis and cell death, cell cycle, and mitochondrial membrane potential were performed in serum-starved THP-1 cells without PMP co-incubation. Cells in the exponential growth phase were washed, resuspended, and kept for 48 h in IMDM before further analysis, as described in the separate sections. For measurement of proliferation rate, cells were resuspended at a concentration of 4 × 10^5^ per mL IMDM, with or without 10% FBS, and counted using the flow cytometer after 24, 48, and 72 h.

### 4.17. Statistical Analysis

Statistical analyses were performed using the IBM SPSS 26 software (IBM Corp., Armonk, NY, USA). Comparison between experimental groups was performed using tests for paired or independent data when appropriate. The data were checked for normality using P-P plots, Shapiro–Wilks test, and Kolmogorov–Smirnov test. A *p* value < 0.05 was considered significant. Mean values are reported with a 95% confidence interval unless otherwise specified. “*n*” denotes technical replicates.

## 5. Conclusions

We show that PMP co-incubation decreases mitochondrial membrane potential, inhibits cell cycle progression, decreases proliferation, and induces differentiation toward macrophages in THP-1 cells. This differentiation effect, combined with decrease in cell activity, may explain the observed protection against daunorubicin-induced cell death, which is also evident in primary AML cells.

Our results warrant further research to explore the in vivo effects of platelet microparticles in AML, both as anti-apoptotic agents, and as modulators of the disease, as they represent possible therapeutic targets through the use of platelet inhibitors.

## Figures and Tables

**Figure 1 cancers-13-01870-f001:**
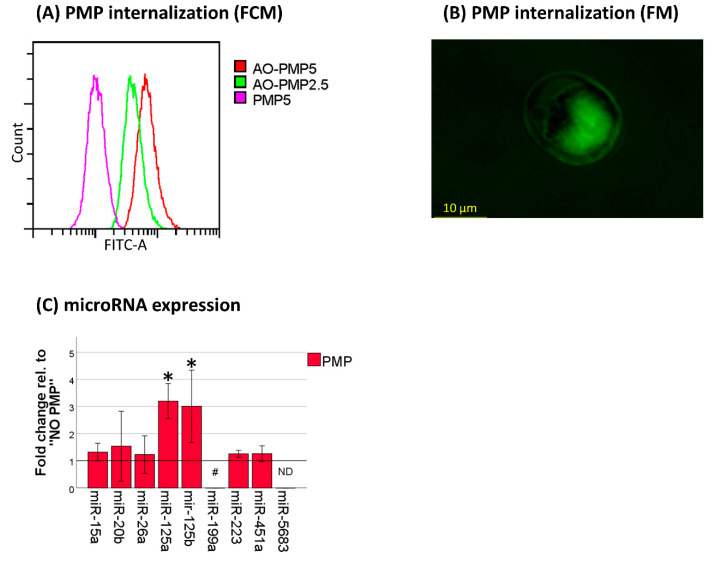
Internalization of platelet microparticles (PMPs) in THP-1 cells after 18 h of co-incubation. (**A**) Transfer of acridine orange from stained PMPs analyzed by flow cytometry (FCM). Histogram plot from a representative experiment (*n* = 2). Number following different PMP groups denotes the final concentration in million PMPs per mL medium. AO, acridine orange. (**B**) Transfer of acridine orange from stained PMPs analyzed by fluorescence microscopy (FM) at 400× magnification. (**C**) Changes in levels of microRNAs (*n* = 3). microRNA data were calculated as fold change from THP-1 without PMP co-incubation, normalized for *BCR*. *p* values were calculated using the one-sample *t* test. * *p* < 0.05. #, fold change was not calculated as levels were undetectable in 2/3 replicates for THP-1 without PMP co-incubation. ND, not detected.

**Figure 2 cancers-13-01870-f002:**
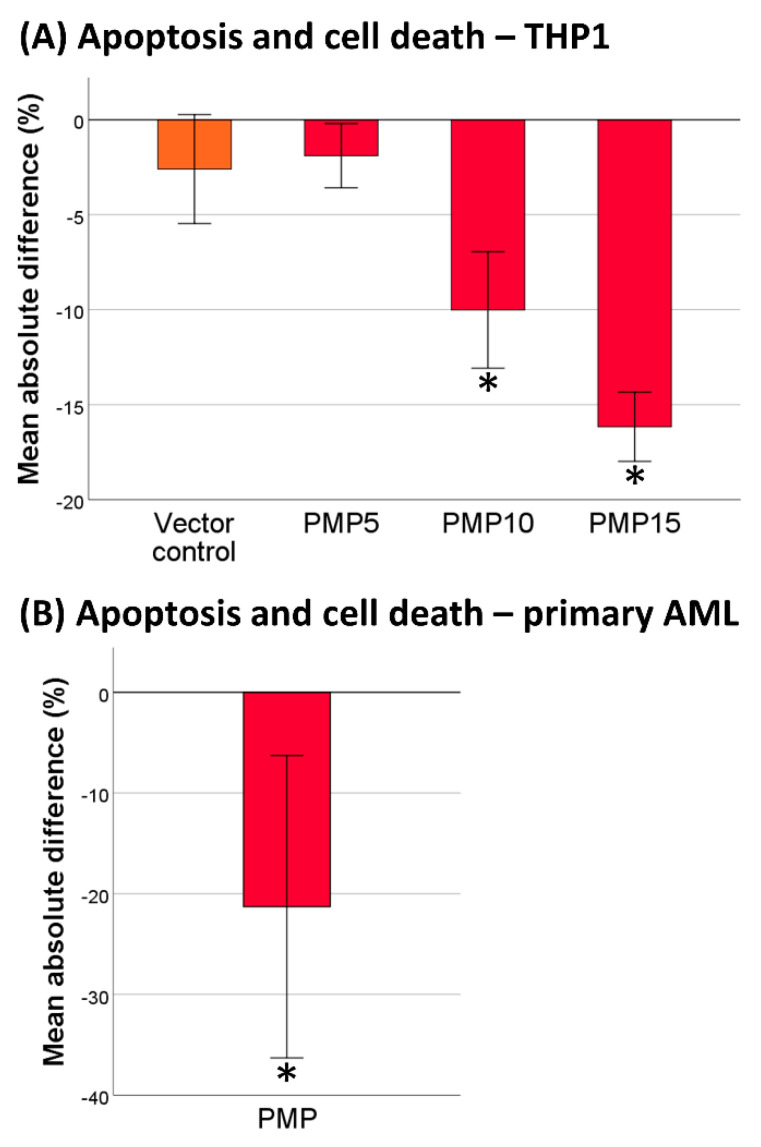
Apoptosis and cell death in THP-1 and primary acute myelogenous leukemia (AML) cells treated with 0.5 µM daunorubicin. (**A**) Difference in frequency of dead and apoptotic cells in THP-1 cells with or without co-incubation with platelet microparticles (PMPs) (*n* = 3). Number following different PMP groups denotes final concentration in million PMPs per mL medium. (**B**) Evaluation of the effect of PMPs on the frequency of dead and apoptotic cells in primary AML cells (nine patient samples, *n* = 4). *p* values were calculated using the one-sample *t* test (test value = 0). * *p* < 0.05.

**Figure 3 cancers-13-01870-f003:**
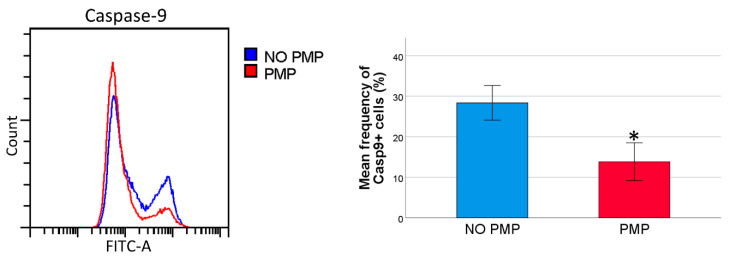
Caspase-9 activation in daunorubicin (DNR)-induced cell death. DNR-treated THP-1 cells were analyzed for caspase-9 activation (*n* = 3). Caspase-9 positive cells were identified as the distinct second peak in the flow histogram. *p* values were calculated using the paired-sample *t* test. * *p* < 0.05.

**Figure 4 cancers-13-01870-f004:**
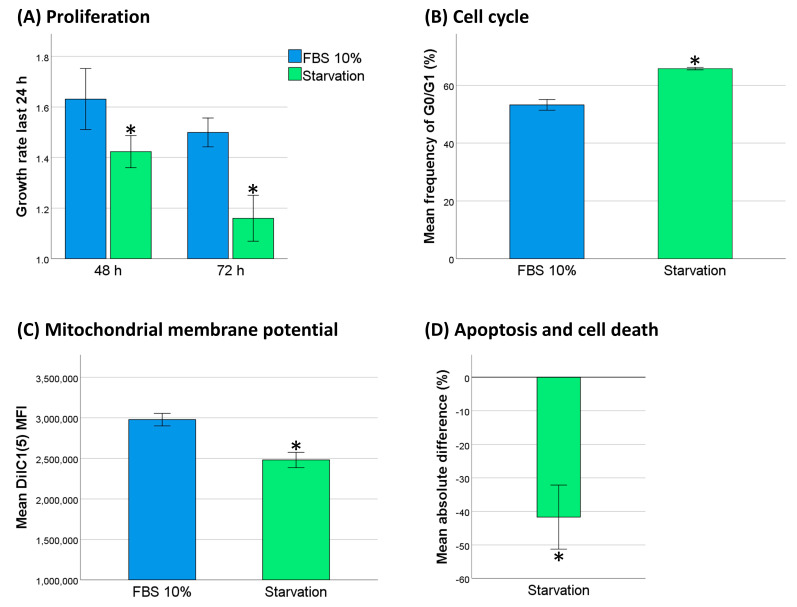
Effects of serum starvation on cell activity and daunorubicin (DNR)-resistance in THP-1 cells. (**A**) Daily proliferation rate analyzed by flow cytometric counting (*n* = 4). (**B**) Cell cycle analysis after 48 h of serum starvation (*n* = 3). Cells in the G0/G1 cell phase were gated. (**C**) Mitochondrial membrane potential after 48 h of serum starvation (*n* = 3), mean fluorescence intensity (MFI) data. (**D**) Difference in DNR-induced cell death and apoptosis after 48 h of serum starvation compared to standard conditions. (**A**–**C**) were compared using the paired-sample *t* test. (**D**) was compared using the one-sample *t* test (test value = 0). * *p* < 0.05.

**Figure 5 cancers-13-01870-f005:**
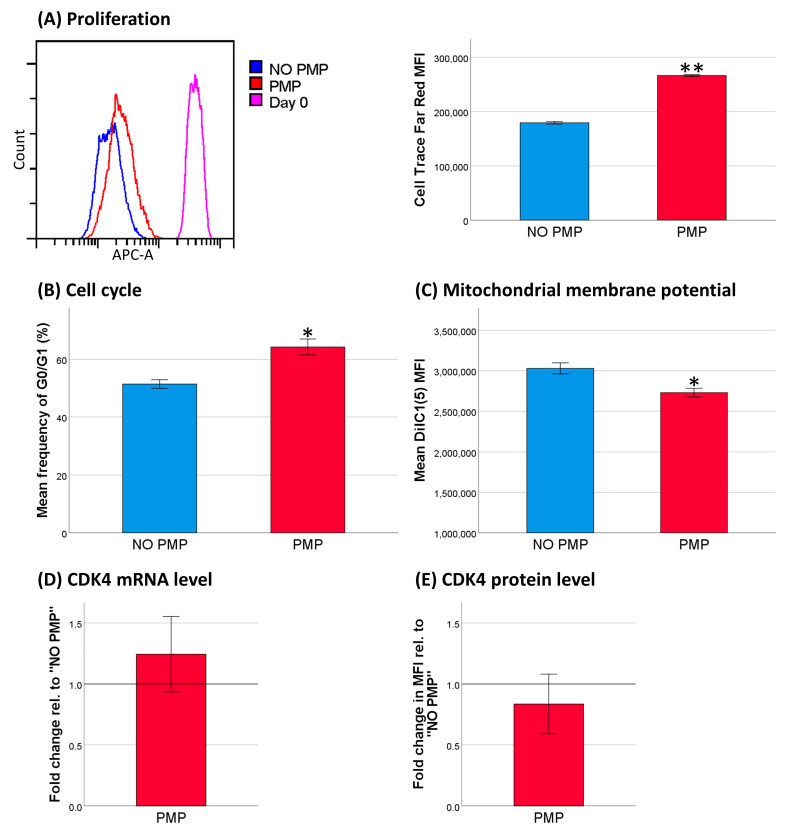
Effects of platelet microparticles (PMPs) on cell activity in THP-1 cells. (**A**) Proliferation analysis by Cell Trace Figure 5. (*n* = 5). (**B**) Cell cycle analysis (*n* = 3). Cells in G0/G1 cell phase were gated. (**C**) Mitochondrial membrane potential (*n* = 4), mean fluorescence intensity (MFI) data. (**D**) CDK4 mRNA levels (*n* = 3). (**E**) CDK4 protein levels (*n* = 5). (**B**–**E**) were analyzed after 24 h of PMP co-incubation. mRNA data are calculated as fold change from THP-1 without PMP co-incubation, normalized for *ACTB*. Protein data are calculated as fold change in MFI from THP-1 without PMP co-incubation. Data were compared using the paired-sample *t* test for data pairs or the one-sample Wilcoxon signed-rank test for ratios. * *p* < 0.05. ** *p* < 0.001.

**Figure 6 cancers-13-01870-f006:**
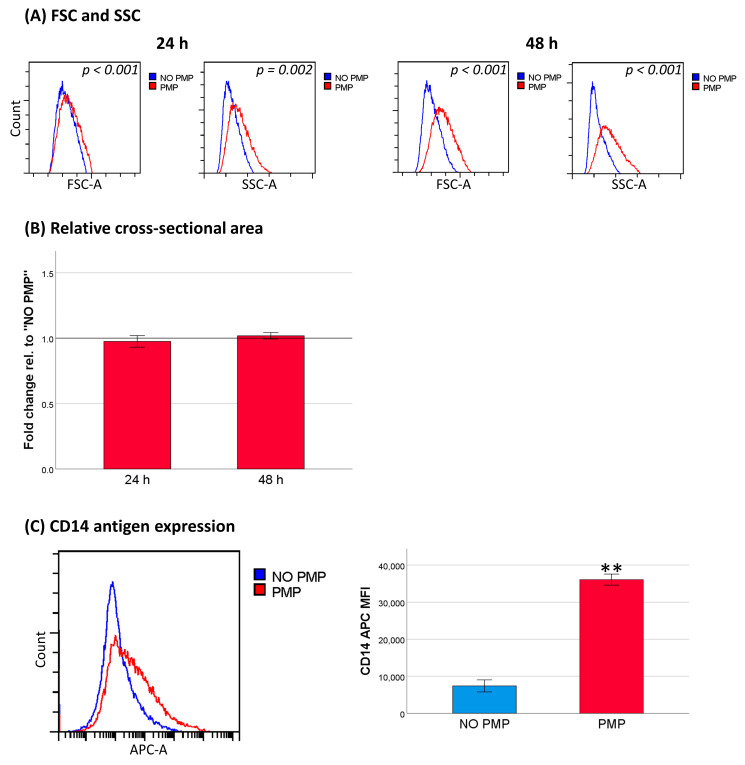
Phenotypical changes induced by platelet microparticles (PMPs) co-incubation. (**A**) Forward scatter (FSC) and side scatter (SSC; *n* = 4). (**B**) Ratio of cell cross-sectional area (*n* = 4). (**C**) CD14 antigen expression after 48 h of culture (*n* = 5). Data were compared using the paired-sample *t* test for data pairs or the one-sample *t* test for ratios. ** *p* < 0.001.

**Table 1 cancers-13-01870-t001:** Characteristics of primary AML patients.

#	Sex	Age	Prev. Myeloid Disease	FAB	Cytogenetics	FLT3	NPM1	CEBPA	CD11b	CD14	CD33	CD34	CD45	CD64	CD117	HLA-DR	L-MPO
1	M	22	No	M5	del(9)	ITD (low ratio)	wt		neg	neg	pos	pos	dim		pos	pos	dim
2	F	60	PV		del(7)	wt	wt	wt	neg	neg		pos	dim		pos	pos	
3	F	56	No	M4	inv(16)	wt	wt		neg	neg	pos	pos	dim	neg	pos	pos	pos
4	M	44	No	M4	inv(16)	wt	wt		neg	neg	neg	neg	dim	dim	neg	pos	
5	F	92	No	M1					neg	neg	dim	neg	dim	dim	pos	pos	pos
6	M	49	No	M4	45, XY	wt	ins	wt	neg	neg		neg	dim	dim	pos	pos	
7	M	76	No	M5	Normal	wt	ins	wt	hetero	hetero		neg	hetero	pos	hetero	pos	
8	F	95	No	M4	Normal	wt	wt	wt				dim	dim		dim		
9	M	29	No	M4	Normal	ITD (high ratio)	wt	wt	neg	neg	neg	pos	dim	neg	neg	pos	pos

# patient number.

## Data Availability

The data presented in this study are available on request from the corresponding author.

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
