# Peer review of "Platelet Microparticles Protect Acute Myelogenous Leukemia Cells against Daunorubicin-Induced Apoptosis"

_cancers, 2021, doi:10.3390/cancers13081870_

Round 1

Reviewer 1 Report

The Authors present a paper titled: " Platelet microparticles protect acute myelogenous leukemia cells against daunorubicin-induced apoptosis" that as to me is very interesting from a biological point of view with possible important clinical impact. The manuscript is well documented and the study well developed. I don't have specific criticisms only minor adjustments.

  1. why Discussion (chapter 3) is before Material and Methods (chapter 4.)?
  2. in the Conclusions please add a sentence underlying the role in the clinical application could have this interesting biological study.

Author Response

Reviewer 1

1. Why Discussion (chapter 3) is before Material and Methods (chapter 4)?

This setup was chosen according the journal’s preference.

2. In the Conclusions please add a sentence underlying the role in the clinical application could have this interesting biological study.

Thank you for the suggestion. It is now implemented in the Conclusions that production of PMPs in AML could potentially be targeted with platelet inhibition (line 455), which is a reference to the last paragraph in the Discussion where this is elaborated more in detail.

Reviewer 2 Report

The manuscript “Platelet microparticles protect acute myelogenous leukemia cells against daunorubicin-induced apoptosis” by Cacic et al analyses the consequences of platelet microparticle (PMP) uptake by THP-1 cells, a monocytic AML cell line, with regard to daunorubicin-induced apoptosis.

While this is a highly interesting and clinically relevant topic, several points should be addressed before being acceptable for publication.

One major concern is the limited novelty of the findings of this study. Furthermore, the analysis of the functional consequences of PMP internalization (microRNA uptake, cell cycle arrest, differentiation), stays rather descriptive and offers little novel mechanistic insight into how exactly uptake of PMPs leads to the observed effects.

  1. The authors measure increase of selected microRNAs, such as miR-125a-5p, miR-125b-5p, and miR-199-5p, in THP-1 cells upon uptake of PMPs. “These findings indicate that microRNAs can be transferred from platelets to THP-1 cells through PMP internalization.” Though likely, experimental evidence for this is actually lacking. Should be toned down a bit.
  2. While the authors write that they identified potential targets of miR-125a-5p and miR-125b-5p, such as BMF and BAK, this is not further analysed. Instead, the authors switch to cell cycle analysis, leaving the reader wondering about the relevance of elevated miR-125a-5p and miR-125b-5p levels. A validation of BMF and BAK as miR-125a-5p/miR-125b-5p targets would be helpful at this point.
  3. The finding that CDK4 levels are not altered does not add any insights into why cell cycle progression is actually attenuated. A more detailed analysis of additional factors would greatly improve the manuscript.

Minor points :

  • 6C Y-axis is not labelled
  • Please check the journal names in the references, e.g. ref. 28. For ref 22, the journal´s name is missing.

Author Response

Reviewer 2

The manuscript “Platelet microparticles protect acute myelogenous leukemia cells against daunorubicin-induced apoptosis” by Cacic et al analyses the consequences of platelet microparticle (PMP) uptake by THP-1 cells, a monocytic AML cell line, with regard to daunorubicin-induced apoptosis.

While this is a highly interesting and clinically relevant topic, several points should be addressed before being acceptable for publication.

One major concern is the limited novelty of the findings of this study. Furthermore, the analysis of the functional consequences of PMP internalization (microRNA uptake, cell cycle arrest, differentiation), stays rather descriptive and offers little novel mechanistic insight into how exactly uptake of PMPs leads to the observed effects.

  1. The authors measure increase of selected microRNAs, such as miR-125a-5p, miR-125b-5p, and miR-199-5p, in THP-1 cells upon uptake of PMPs. “These findings indicate that microRNAs can be transferred from platelets to THP-1 cells through PMP internalization.” Though likely, experimental evidence for this is actually lacking. Should be toned down a bit.

We have specified that the increase in microRNAs is indirect proof of transfer and eliminated phrases indicating that microRNAs are directly transferred (line 86, 98, and 211).

  1. While the authors write that they identified potential targets of miR-125a-5p and miR-125b-5p, such as BMF and BAK, this is not further analysed. Instead, the authors switch to cell cycle analysis, leaving the reader wondering about the relevance of elevated miR-125a-5p and miR-125b-5p levels. A validation of BMF and BAK as miR-125a-5p/miR-125b-5p targets would be helpful at this point.

The relevance of BMF and BAK being potential targets is based on the knowledge that both of these proteins are members of the proapoptotic BCL2 family proteins which activate the caspase-9 pathway. This is tested and found to be less activated after co-incubation with PMPs. Then, focus is shifted towards changes in cell activity because, as stated in the discussion, these microRNAs are also known to decrease cell activity in multiple cell lines. Although BMF and BAK are not tested as targets for the microRNAs, transfer of the microRNAs could still be a potential mechanism for the chemoprotective effect.

  1. The finding that CDK4 levels are not altered does not add any insights into why cell cycle progression is actually attenuated. A more detailed analysis of additional factors would greatly improve the manuscript.

 We agree. We do have data with another cell cycle regulator, E2F2, also a predicted target for miR-125a/b. We found that mRNA level was unchanged with PMP co-incubation, but surprisingly protein level was slightly increased about 20%. Whether this discrepancy is stochastic and mere a result of the limitations of qPCR for detecting small differences or other technical issues is unknown. There are also discrepancies in the literature regarding the function of E2F2, as it is described as both a positive and negative regulator of cell cycle [1, 2], and the function may shift as a result of differentiation [3]. This could be a possible explanation for the PMP effect on proliferation, but little data are available on the role of E2F2 in leukemic cells, and if it were so one would except a bigger difference. In addition, an increase of E2F2 protein could not have been explained by transfer of miR-125a/b, as one would expect a decrease in protein level if E2F2 mRNA was a target. There are several papers that have investigated platelet proteomics, and to the authors’ knowledge it is very unlikely that E2F2 is transferred directly from PMPs, so the effect would have to be indirect. Since there are several uncertainties regarding these findings, the data were excluded. Still, further investigation of other cell cycle regulators is an excellent idea, but it is hard to pinpoint exactly which to implement. 

Minor points :

  • 6C Y-axis is not labelled

Figure 6C is a flow cytometry histogram where the Y-axis represent “count”. This is also true for Figure 1A, Figure 3, Figure 5A, and Figure 6A. We have now labeled them accordingly. 

  • Please check the journal names in the references, e.g. ref. 28. For ref 22, the journal´s name is missing.

All references are imported from pubmed.gov and inserted using EndNote software, and now we have used the output style recommended by MDPI and the journal. But as remarked, for reference 22 journal name was missing and this is now corrected.

 References

  1. Infante, A., et al., E2F2 represses cell cycle regulators to maintain quiescence. Cell Cycle, 2008. 7(24): p. 3915-27.
  2. Laresgoiti, U., et al., E2F2 and CREB cooperatively regulate transcriptional activity of cell cycle genes. Nucleic Acids Res, 2013. 41(22): p. 10185-98.
  3. Chong, J.L., et al., E2f1-3 switch from activators in progenitor cells to repressors in differentiating cells. Nature, 2009. 462(7275): p. 930-4.

Reviewer 3 Report

Here Cacic and colleagues describe the role of platelet microparticles protecting AML against daunorubicin, potentially through differentiation. 

My major comment is that almost the entire work was done with one cell line THP-1, except for Figure 2B showing primary AML cells. However, in this experiment no control was used as was done for the THP-1 cells. From the methods it appears they might have been compared to THP-1 cells?  Also, the table and legend say 9 AML samples and n=4, but just one graph is shown. Is this data for all 9 samples with 4 replicates = 36 assays? Can the authors explain? 

THP-1 cells are monocytic leukemia cells and known for phagocytosis and ability to differentiate. Do the PMPs actually get into patient leukemia cells or other leukemia cell lines? Can the authors check the differentiation markers after PMP treatment of primary human AML cells? 

Author Response

Reviewer 3

Here Cacic and colleagues describe the role of platelet microparticles protecting AML against daunorubicin, potentially through differentiation. 

  1. My major comment is that almost the entire work was done with one cell line THP-1, except for Figure 2B showing primary AML cells. However, in this experiment no control was used as was done for the THP-1 cells. From the methods it appears they might have been compared to THP-1 cells?  Also, the table and legend say 9 AML samples and n=4, but just one graph is shown. Is this data for all 9 samples with 4 replicates = 36 assays? Can the authors explain?

Both diagrams in Figure 2 represents the difference of dead and apoptotic cells from cells treated with platelet microparticles compared to the same cells (cell line or patient samples) in a “NO PMP” setting (not treated with platelet microparticles). This means that there are specific controls for THP-1 and each of the nine donor cells. Frequency of dead and apoptotic cells treated with PMPs are subtracted with frequency of dead and apoptotic cells from the same cells without PMP co-incubation. To clarify, daunorubicin was added in all cultures that the presented data are derived from. For the experiments with primary cells, THP-1 cells were also cultured using the medium master mix for primary AML cells as a quality control to detect potential false negative results. In the original manuscript we called these experiments “positive controls”, but this is now changed to “quality controls” (line 134-137 and 365-371). These data were also used to analyze inter-batch variance, which was not significant. To clarify, these are separate experiments from Figure 2A, and are not used to calculate chemoprotective effect of PMPs in primary AML cells.

 There are four technical replicates for every patient sample (with and without PMPs). The average value of the replicates is used in Figure 2B where all patient data are compared so that here, “n” is actually nine (but in the manuscript “n” is defined as technical replicates, which is why it is specified as “nine patient samples, n=4”). In Figure S1 intraindividual effect of platelet microparticles is assessed, where n=4, just comparing the effect of PMPs within separate donors.

  1. THP-1 cells are monocytic leukemia cells and known for phagocytosis and ability to differentiate. Do the PMPs actually get into patient leukemia cells or other leukemia cell lines? Can the authors check the differentiation markers after PMP treatment of primary human AML cells? 

 As primary AML cells are a limited resource, we could not do elaborate analysis of cell activity and differentiation. We did try to use the Cell Trace assay to assess the effect of PMPs on proliferation of the primary AML cells. Although there were indications that proliferation was inhibited, these data were excluded as the dye itself was too toxic for the cultures to sustain a sensible proliferation in brittle cryoperserved AML cells. We have done thorough testing with the assay with THP-1 cells, and toxicity is negligible when using THP-1 cells in exponential growth phase.

 PMP uptake is proven in other AML cell lines, and also primary leukemia patient cells:

 Kailashiiya et al. show uptake of PMPs through transfer of several fluorescent compounds in HL60 and K562 cell lines as well as patient samples from AML, ALL, and CLL [1].

 Qu et al. show uptake of PMPs through transfer of PKH67 label in K562, MEG01, and UT-7 cell lines. Although not as relevant, they also show PMP uptake in hematopoietic stem and progenitor cells from human cord blood samples [2].

References

  1. Kailashiya, J., V. Gupta, and D. Dash, Engineered human platelet-derived microparticles as natural vectors for targeted drug delivery. Oncotarget, 2019. 10(56): p. 5835-5846.
  2. Qu, M., et al., Platelet-derived microparticles enhance megakaryocyte differentiation and platelet generation via miR-1915-3p. Nat Commun, 2020. 11(1): p. 4964.

Round 2

Reviewer 2 Report

All minor points have been addressed.

Author Response

As far as I understand there are no further comments needing reply.